# Do Molecular Fingerprints Identify Diverse Active Drugs in Large-Scale Virtual Screening? (No)

**DOI:** 10.3390/ph17080992

**Published:** 2024-07-26

**Authors:** Vishwesh Venkatraman, Jeremiah Gaiser, Daphne Demekas, Amitava Roy, Rui Xiong, Travis J. Wheeler

**Affiliations:** 1Department of Chemistry, Norwegian University of Science and Technology, 7034 Trondheim, Norway; 2School of Information, University of Arizona, Tucson, AZ 85721, USA; 3R. Ken Coit College Pharmacy, University of Arizona, Tucson, AZ 85721, USA; 4Rocky Mountain Laboratories, Bioinformatics and Computational Biosciences Branch, Office of Cyber Infrastructure and Computational Biology, National Institute of Allergy and Infectious Diseases, National Institutes of Health, Hamilton, MT 59840, USA; amitava.roy@umontana.edu; 5Department of Biomedical and Pharmaceutical Sciences, University of Montana, Missoula, MT 59812, USA; 6Department of Pharmacology & Toxicology, University of Arizona, Tucson, AZ 85721, USA

**Keywords:** molecular fingerprints, drug screening, similarity searching

## Abstract

Computational approaches for small-molecule drug discovery now regularly scale to the consideration of libraries containing billions of candidate small molecules. One promising approach to increased the speed of evaluating billion-molecule libraries is to develop succinct representations of each molecule that enable the rapid identification of molecules with similar properties. Molecular fingerprints are thought to provide a mechanism for producing such representations. Here, we explore the utility of commonly used fingerprints in the context of predicting similar molecular activity. We show that fingerprint similarity provides little discriminative power between active and inactive molecules for a target protein based on a known active—while they may sometimes provide some enrichment for active molecules in a drug screen, a screened data set will still be dominated by inactive molecules. We also demonstrate that high-similarity actives appear to share a scaffold with the query active, meaning that they could more easily be identified by structural enumeration. Furthermore, even when limited to only active molecules, fingerprint similarity values do not correlate with compound potency. In sum, these results highlight the need for a new wave of molecular representations that will improve the capacity to detect biologically active molecules based on their similarity to other such molecules.

## 1. Introduction

Methods for the computational identification of small molecules likely to bind to a drug target (virtual screening) are increasingly intended to explore a space of billions of candidate molecules [1,2,3,4]. One strategy for exploring this massive molecular search space is to begin with a collection of molecules known or presumed to be biologically active (hereafter referred to as *actives*). Those actives can then be used as the basis of a rapid search among billions of candidates [5] for other molecules expected to demonstrate similar activity [6,7].

The notion that small molecules with similar structures are likely to share biological properties, coined the *similar property principle* (SPP) [8,9], is central to this sort of search strategy. The SPP is simple and intuitive, and has served as the basis for predictions of biological activity [10,11], toxicity [12,13,14], aqueous solubility [14,15] (log*S*), and partition coefficient [16] (log*P*). However, structural similarity may not necessarily reflect similarity in biological activity [17], a concept popularly addressed by the notion of an Activity Cliff [18,19,20]. Furthermore, the proper definition of structural similarity depends on the context. For example, the quantitative structure–activity relationship focuses on the similarity between the local structural features of two molecules, while similarity in biological activity typically depends on having more global features of the molecules [21,22,23] (though even these notions of *local* and *global* similarity are also not well defined).

The most common way to quantify the structural similarity of two small molecules begins with the calculation of a so-called *molecular fingerprint*, a binary or count vector that encodes structural and often chemical features [24,25,26]. Such a fingerprint is computed for each molecule, then the fingerprints of molecules are compared for overlap to approximately assess molecular similarity, motivated by the notion that this measure of fingerprint similarity will correlate with the similarity at the level of biological activity. Fingerprint similarity has been used to effectively estimate log*S* and log*P* values [27], likely because these values can largely be approximated from the small molecule itself without explicitly considering interacting partners.

Other molecular properties involve a dependency on context, placing greater strain on the utility of the SPP. For example, the biological activity of a small molecule depends on the interaction between that molecule and the target protein binding region. Such binding regions (or pockets) vary between different proteins, and therefore impose strong context dependence in biological interactions. Consequently, small molecule ligand-based fingerprint similarity may not be sufficient to capture the wide spectrum of similarities in biological activities. Similarly, the toxicity of a small molecule depends on the molecule’s interaction with multiple proteins, suggesting a limit to the inference power provided by similarity at the level of molecular fingerprints.

Despite previous demonstrations of the limitations of using SPP to predict similarity in biological activity [28,29], the technique is heavily used in drug development. This is especially true in fingerprint-based virtual screening (VS), in part due to the computational simplicity and speed of searching the vast chemical space of small molecules [2,30,31,32]. A variety of molecular fingerprints have been devised for use in ligand-based virtual screening (LBVS), to aid in identifying biologically active small molecules [32,33] (hereafter referred to as actives), from within a library of small molecules. LBVS begins with a small number of known/predicted actives (queries) for a target protein pocket, and explores a library of other small molecules, seeking a set that is expected to also be active. This expectation is based on the SPP, so that LBVS seeks molecules with high fingerprint similarity to one of the queries, under the assumption that fingerprint similarity to known actives will generally assign a higher ranking to actives than to non-actives (decoys). Martin et al. [28] have highlighted a disconnect between empirical and computational perception of similarity, and suggested resulting limitations in the utility of LBVS for drug binding predictions. Here, we explore the shortcomings of simplistic molecular fingerprints in the context of LBVS, and demonstrate that all commonly used fingerprint methods fail to sufficiently enrich for binding activity in a library of mostly decoy molecules. We are not oblivious to a prevailing perception that fingerprint similarity is suggestive of similarity in drug interaction potential, and hope that this manuscript contributes to a better understanding of the limitations of fingerprint similarity in the context of large-scale drug screening.

## 2. Results

To gain insight into the utility of various fingerprinting strategies for billion-scale virtual drug screening, we explored the capacity of fingerprint similarity to extract a small set of candidates that is highly enriched for molecules with activity similar to the seed query molecules. First, we computed measures of enrichment for 32 fingerprints on four benchmark data sets, presenting both classical ROC AUC calculations and our new decoy retention factor (DRF) scores. We then explored the distributions of fingerprint similarity scores across a variety of target molecules, and show that the score distributions for actives and decoys are not sufficiently separable for billion-scale search. We also found that, where fingerprints did occasionally find good matches among actives, the “new” actives almost always shared a scaffold with the query active (meaning that they would be found by more traditional shape exploration methods). We further considered whether there is a correlation between compound potency and active–active similarity scores, and found that there is not. Finally, we used a data set containing more than 300,000 experimentally confirmed inactive compounds, and found that fingerprint similarity to an active molecule does not enable discrimination between actives and inactive. In total, these results indicate that fingerprint similarity is not a reliable proxy for likely similar binding activity or particularly for discovering diverse active structures.

### 2.1. Enrichment for Active Molecules

To assess the utility of fingerprinting strategies for selecting compounds with similar expected activity, we computed the similarities of all compounds to a query active molecule, and tested whether active molecules tend to be more similar to other actives than to decoys. Specifically, for each target protein, we computed the fingerprints of each molecule associated with that target protein. Then, for each active compound, we computed the similarity of its fingerprint to each of the other compounds (actives and decoys) affiliated with that target. The union of these active/decoy distance calculations was merged and sorted by similarity, enabling calculation of DRF${\;\;}_{0.1}$ (see Section 4) and ROC AUC for each fingerprint–target pair. Then, for each fingerprint–target pair, the mean value (AUC or DRF${\;\;}_{0.1}$) was computed over all target proteins in the corresponding benchmark.

Table 1 presents the resulting enrichment values on each benchmark data set. The performance of all fingerprints is poor for both the MUV and LIT-PCBA data sets, with AUC values generally <0.6, and DRF${\;\;}_{0.1}$ values close to 1.0 (indicating small enrichment of actives relative to decoys). Performance is somewhat better on DEKOIS and DUD-E, but not particularly strong, and is offset by concerns of benchmark bias highlighted elsewhere, such as artificial enrichment [34,35,36,37,38,39] (enrichment due to bias in the actives or decoys), analogue bias (limited diversity of the active molecules), and false negative bias (risk of active compounds being present in the decoy set), all of which can cause misleading VS results [37,39,40]. Table 1 also provides a summary of the VS performances obtained for the fingerprint types (substructure, circular, path, text, pharmacophore). No particular fingerprint strategy appears to be better suited to the problem of virtual screening. Among all tests, the best observed DRF${\;\;}_{0.1}$ value was 0.09, equating to a roughly 11-fold enrichment in actives versus inactives; in a screen of a billion molecules, this means that roughly 9 million inactive molecules are expected to show similarity at the score threshold where 10% of actives are retained).

Most circular and path-based fingerprints employ a standard length of 1024 bits. O’Boyle and Sayle [41] suggested that increasing the bit-vector length from 1024 to 16,384 can improve VS performance, though at a cost of space and run time for comparison. We evaluated the utility of longer fingerprints for the MUV and LIT-PCBA data sets, and found that longer fingerprints yield little to no gain in efficacy (see Appendix A).

### 2.2. Tanimoto Similarity Distributions Are Generally Indistinguishable

To explore the distribution of similarities between actives and decoys, we computed Tanimoto coefficients for active–active and active–decoy molecule pairs in the DEKOIS data set. For each target protein in DEKOIS [42], we randomly selected an active molecule, and computed the molecular fingerprint Tanimoto similarity to all other actives and decoys for that target. Figure 1 shows the resulting score distributions for 32 fingerprints. The distributions of active–active (blue) and active–decoy (red) Tanimoto values substantially overlap, so that the vast majority of actives fall into a score range shared by most decoys. (Note: a single active molecule was used as query in order to replicate a common use case for fingerprints in virtual screening. Similarly overlapping distributions are observed when merging the results of using all DEKOIS actives as query seeds, but with a more complex visual landscape due to a mixture of similarity distributions.)

Most of the fingerprints in Figure 1 present a thin high-Tanimoto tail for actives (blue) that is not seen for decoys (red), suggesting that perhaps a small fraction of actives could be discriminated from decoys by establishing a high score threshold—in other words, it may be possible to select a threshold that delineates regions of early enrichment. However, consider the ECFP2 fingerprint, which shows an apparently compelling right tail in the active–active plot (blue), such that it appears to be reasonable to establish a Tanimoto cutoff of 0.5. In DEKOIS, there are 423 active matches to active queries above this threshold. Though the right tail of the active–decoy distribution (red) is imperceptible in this plot, it still contains ∼0.0064% of the decoys. Extrapolating to a library of 3.7 billion candidates, as we used in Venkatraman et al. [2], we expect to see ∼23.7M decoys with Tanimoto ≥ 0.5, so that the active-to-decoy ratio is ∼1:56,000. Setting the Tanimoto threshold to 0.75 leads to an expected ratio of ∼1:68,000 (57 actives to ∼3.9 M expected decoys). This is likely an overly pessimistic view of the decoy ratio risk, since the compounds in the decoy set are intended to be more similar to the actives than would be a random molecule from a billion-scale library. Even so, it highlights the fact that even small false discovery rates can produce an overwhelmingly large number of false matches when the target base is sufficiently large. Moreover, see the next section for an exploration of the expected novelty of these high-scoring matches.

### 2.3. Tanimoto Score Distributions, and Their Utility in Scaffold Hopping

The LIT-PCBA data set [43] contains binding activity data for 15 proteins, each with 13 to 1563 confirmed active molecules and 3345 to 358,757 confirmed inactives/decoys (see Table 2). We computed the ECFP4 fingerprint for each of the 2.6 M active and inactive small molecules. For each protein, we first selected the molecule with best affinity to serve as the ‘active query’, then computed the Tanomito similarity measure ($T_{c}$) between seed and all other molecules (actives and decoys) for that protein. Figure 2 presents $T_{c}$ distributions. Each plot shows, for a single protein, the fraction of actives (blue) and decoys (red) that are recovered (Y-axis) as a function of $T_{c}$ (X-axis). One important observation is that *for most proteins*, the molecule with the highest similarity (active or decoy) to the active query has a $T_{c}<0.5$, which is below commonly used thresholds for expected shared bioactivity [44,45,46]. In addition, for most proteins, $T_{c}$ distributions are indistinguishable, though three proteins (GBA, OPRK1, and PPARG) demonstrate early enrichment for actives. We manually inspected all high-scoring actives ($T_{c}>0.5$) for these three enriched proteins (see Figure 3 and Appendix A), and observed that high-scoring actives were almost entirely bioisosteres, with easily predictable results of exchanging an atom or atom-group with a similar atom or group. The lack of scaffold diversity among matches with scores discernible from high-volume decoy noise casts doubt on the utility of fingerprint similarity for exploring novel candidate drug spaces.

### 2.4. Evaluation on a Target with Many Validated Inactive Molecules

The previous experiments depend on benchmarks containing computationally identified decoys that almost entirely have not been experimentally validated as inactive. The MMV St. Jude malaria data set [47] provides an alternative perspective on the utility of fingerprint similarity for activity prediction in the context of verified decoys. It contains a set of 305,810 compounds that were assayed for malaria blood stage inhibitory activity. Among these molecules, 2507 were classified as active, while the remaining 303,303 compounds were classified as inactive.

For each active molecule, we computed Tanimoto similarity to every other active and all inactives. Figure 4 shows bar plots for each fingerprint, with each plot showing the fraction of inactives (red) and actives (blue) with Tanimoto similarity values $T_{ab}\geq c$ for values of *c* = (0.1, 0.2, …, 0.9) and 0.99. In general, the remaining fraction of actives only slightly exceeds the remaining fraction of inactives, indicating minimal enrichment of actives at increased Tanimoto similarity values. MAP4 shows an apparent relative abundance of actives, but note enrichment is still only ∼10-fold; also note that <1.5% of actives show Tanimoto similarity >0.1 to another active, raising concerns about how to establish meaningful MAP4 thresholds.

One important caveat for this data set is that the target binding pocket of each active is unknown, so that it is possible that some actives target one pocket while other actives target another pocket. Even so, the lack of visible signal of predictive enrichment from fingerprint similarity is notable.

### 2.5. Fingerprint Similarity Values Do Not Correlate with Compound Potency

The previous sections demonstrate that fingerprint similarity has limited utility in discriminating active molecules from decoys. An alternative use of fingerprints could be to take a set of candidates that have already (somehow) been highly enriched for active compounds, and rank them according to expected potency. For the LIT-PCBA data set, these potency values for drug–target pairs were retrieved from PubChem [48]: AC${\;\;}_{50}$ (“Half-maximal Activity Concentration”, defined as the concentration at which the compound exhibits 50% of maximal activity).

For each target protein in the LIT-PCBA data set, we selected the most potent active molecule, and computed fingerprint similarities for all other actives for the corresponding target. We evaluated the correlation of fingerprint similarity value to observed AC${\;\;}_{50}$ by computing the Kendall rank correlation [49]. Figure 5 presents a heatmap of these correlation values (τ) for each fingerprint across 15 protein targets, and demonstrates that all fingerprints exhibit poor correlation, with values ranging between −0.53 to 0.54, and generally only slightly higher than zero. This indicates that the fingerprints evaluated are unlikely to yield a ranked set of enriched highly potent compounds, in agreement with the observations of Vogt and Bajorath [50].

To give insight into these summary statistics represented by the heatmap squares in Figure 5, we prepared scatter plots corresponding to three of the target/fingerprint pairings (Figure 6). The middle plot shows the relationship between fingerprint similarity and AC${\;\;}_{50}$ values for the target/fingerprint pair with median Kendall correlation (target = VDR, fingerprint = SIGNATURE), and is representative of most of the target/fingerprint pairs; it shows essentially no correlation between fingerprint similarity and AC${\;\;}_{50}$ values (τ=0.01). The first and last scatter plots show fingerprint similarity and AC${\;\;}_{50}$ values for the target/fingerprint with the highest (ADRB2, FFCP0) and lowest (PPARG, ASP) correlation values. Note that Kendall rank correlation values for FCFP0 with targets other than ADRB2 (τ=0.54) vary from −0.30 to 0.05 and for ASP with targets other than PPARG (τ=−0.53) from −0.23 to 0.18. Though some protein/fingerprint combinations show apparent correlation between fingerprint similarity and AC${\;\;}_{50}$, most show a near total lack of correlation; deviations (positive or negative correlation) are observed in proteins with a small number of actives, which are thus more prone to stochastic or biased observations of correlation.

## 3. Discussion

Computational methods hold the promise of expanding the diversity of small molecule drug candidate targeting for specified protein binding pockets. To fulfill this promise, these methods must be supported by succinct molecular representation schemes that enable the rapid identification of molecules that are functionally similar to known or suspected active molecules while presenting non-trivial molecular similarity. The results of this study demonstrate that molecular fingerprints, and specifically the measurement of molecular similarity based on those fingerprints, are not generally effective at discriminating molecules with similar binding activity from those with dissimilar activity; furthermore, we have shown that where some early enrichment is observed, it tends to highlight candidate molecules that would arise from a simple structural perturbation. In addition to explicitly highlighting the weakness of molecular fingerprints for drug similarity assessment, the results also suggest that fingerprints are unlikely to be useful for the calculation of classification confidence or applicability domain [51]. In total, these results suggests that the task of the rapid evaluation of molecular similarity must move beyond the fingerprint representation of molecules.

The binding of a ligand to a target is a function of the target protein, the ligand, and other nearby molecules. In complicated cases, the role of nearby molecules can include cases where cooperation between multiple molecules is required for a binding event to occur. The ensemble properties of all the participant molecules should be considered to define similarity in the context of biological activity related to protein–ligand binding—a task difficult to achieve by focusing on ligand structure alone. The difficulty is poignantly highlighted by the fact that a simple atom count-based descriptor performs slightly worse than some fingerprint-based descriptors in defining similarity in biological activity [52]. When multiple active ligands are present for a target, with diverse chemical and structural properties differentiating consequential features from inconsequential ones in a binding event, fingerprint-based methods can possibly identify a novel active compound for that target. Multiple recent studies have shown success following that approach [53,54,55]. However, in most practical cases, such information is not available. It is important to understand the limitation of defining “similarity” based on fingerprints in the context of biological activity and use such methods in conjunction with other orthogonal methods for a successful design of VS.

In recent years, deep learning strategies have entered the drug discovery toolkit (e.g., [56,57,58]), but these have not yet solved the problem of rapid virtual screening. Though the path forward is not clear, we suggest that it is vital that molecules be represented in such a way that the potential *context* of the molecule (i.e., information about the potential binding target) can be considered when evaluating the similarity of molecules. We suspect that future successful strategies will emphasize the surface properties of the small molecule [59,60], and will represent both the compound and the target protein not as a monolith, but as a collection of surface patches [61,62,63]. These, we believe, will enable a more context-dependent emphasis on features of importance to particular interactions, without interference from unimportant features.

## 4. Materials and Methods

### 4.1. Fingerprint Representations

The palette of fingerprints evaluated in this study (Table 3), can be broadly classified into those based on (i) path, (ii) circular features, (iii) pharmacophore, and (iv) pre-defined generic substructures/keys [64]. Circular and path-based fingerprints are generated using an exhaustive enumeration of (linear/circular) fragments up to a given size/radius, which are then hashed into a fixed-length bit vector. The SIGNATURE descriptor [65] generates explicitly defined substructures, which are mapped to numerical identifiers (no hashing involved). The LINGO fingerprint [16] works directly with the SMILES strings (rather than the usual dependence on a molecular graph representation), by fragmenting the strings into overlapping substrings.

All fingerprints were generated using open-source software. Routines in the RDKit [79] library were used to compute the AVALON, ERG, RDK5, RDK6, RDK7, MHFP, and TT fingerprints. FP2, FP3 and FP4 fingerprint similarities were calculated directly using the OpenBabel toolbox [72]. The other fingerprints were calculated using custom software that makes use of the *jCompoundMapper* [67] and Chemistry Development Kit [70] libraries.

Although a number of similarity metrics have been used [80], the most common measure of fingerprint similarity is the Tanimoto coefficient [81]: $T_{ab}=|F_{a}\cap F_{b}\left|/\right|F_{a}\cup F_{b}|$, where $F_{a}$ and $F_{b}$ are the fingerprints of molecules *a* and *b*, respectively—this is equivalent to the Jaccard index over fingerprint bit vectors. Tanimoto coefficient values range between 1 (identical fingerprints, though not necessarily identical compounds) and 0 (disjoint fingerprints).

### 4.2. Benchmarking Data Sets

Numerous benchmarking data sets have been developed over the years to evaluate VS methods [39,82]. Each data set contains a set of active compounds (with known/documented activity for the target of interest) and a corresponding set of inactives/decoys. While the definition of actives is consistent, there is some variance in the question of what should be considered a ‘decoy’. Some benchmarks include only confirmed inactive molecules, while others add compounds presumed to be non-binding [39,83,84]. Data set composition can impact VS evaluation, such that both the artificial under- and over-estimation of enrichment have been well documented [39,85,86]. To account for benchmark-specific biases and error profiles, we opted to explore fingerprint efficacy across the full set of benchmarks. This evaluation pool provides a diverse perspective on the performance of molecular fingerprints, indicating that limited ability to differentiate decoys from active molecules is not simply due to a specific design flaw found in a single benchmark.

In this study, we employ four different VS data sets to explore the utility of molecular fingerprinting strategies for prediction of similar activity. These data sets are briefly summarized in Table 4 and described below.

**DUD-E:** Directory of Useful Decoys, Enhanced [89]: DUD-E is a widely used data set for VS benchmarking, containing data for 102 protein targets. Each target is represented by an average of ∼224 active ligands, ∼90 experimental decoys, and ∼14,000 computational decoys. Compounds are considered active based on a 1 μM experimental affinity cutoff, and experimental decoys are ligands with no measurable affinity up to 30 μM. Computational decoy ligands are selected from ZINC [83] to have 50 physical properties (rotatable bonds, hydrogen bond acceptors/donors, molecular weight, logP, net charge) similar to the actives, but with low fingerprint (Daylight [90]) Tanimoto coefficient $T_{ab}<0.5$. (Note: this means that computational decoys are, by construction, expected to have low fingerprint similarity).**MUV:** Maximum Unbiased Validation [91]: MUV data sets are based on bioactivity data available in PubChem [48]. This benchmark consists of sets of 30 actives (taken from confirmation assays) and 15,000 decoys (drawn from corresponding primary screens) for each of the 17 targets. The goal of the experimental design is to obtain an optimal spread of actives in the chemical space of the decoys. Since the data are taken from high-throughput screening assays that can be affected by experimental noise and artifacts (caused by unspecific activity of chemical compounds), an assay filter is applied to remove compounds interfering with optical detection methods (autofluorescence and luciferase inhibition) and potential aggregators.**DEKOIS:** The Demanding Evaluation Kits for Objective In silico Screening (DEKOIS) [42] benchmark is based on BindingDB [92] bioactivity data ($K_{i}$, $K_{d}$, or $IC_{50}$ values). The DEKOIS data set is derived from a set of 15 million molecules randomly selected from ZINC, which are divided into 10,752 bins based on their molecular weight (12 bins), octanol–water partition coefficient (8 bins), number of hydrogen bond acceptors (4 bins), number of hydrogen bond donors (4 bins), and number of rotatable bonds (7 bins). Active ligands are also placed into these pre-defined bins. For each active ligand, 1500 decoys are sampled from the active’s bin (or neighboring bins, if necessary). These are further refined to a final set of 30 structurally diverse decoys per active. The DEKOIS data set includes 81 protein targets found in the DUD-E data set.**LIT-PCBA:** The LIT-PCBA benchmark [43] is a curated subset of the PubChem BioAssay database [48], containing data from experiments where more than 10,000 chemicals were screened against a single protein target, and dose–response curves identified at least 50 actives. Active ligands identified in a bioassay experiment are not guaranteed to bind to the same pocket of the target protein; to overcome this concern, LIT-PCBA includes only targets with representative ligand-bound structures present in the PDB, such that the PDB ligands share the same phenotype or function as the true active ligands from the bioassay experiments. The LIT-PCBA data set was further refined to contain only targets for which at least one of the VS methods (2D fingerprint similarity, 3D shape similarity, and molecular docking) achieved an enrichment in true positives (i.e., the most challenging protein targets have been removed, so that enrichment results are, by design, expected to show good results). Targets in the LIT-PCBA have a variable active to decoy ratio that ranges from as low as 1:20 to 1:19,000.

### 4.3. Virtual Screening Evaluation

A common measure for the efficacy of a method’s discriminatory power depends on the receiver operating characteristic (ROC) curve, which plots the sensitivity of a method as a function of false labels [93]. If a classification method assigns better scores to all true matches (actives) than to any false matches (decoys), then the area under that curve (AUC) will be 1. A random classifier will have an AUC of 0.5.

AUC provides a measure of the sensitivity/specificity trade-off across the full sensitivity range, but medicinal chemists are typically more interested in the early recognition of active molecules [32,94], since there is little actionable value to gain in the recall of true actives buried among thousands (or more) of decoys. As an example, consider an imaginary method that assigns the highest scores to 10% of all active molecules, then afterwards loses discriminative power and assigns essentially random scores to all remaining molecules (actives and decoys). The AUC for such a method would be not particularly good (roughly 0.6), even though the early enrichment (ranking 10% of actives with a superior score to all decoys) provides some experimental utility.

To address this shortcoming of ROC AUC, a number of other metrics have been devised to assess early enrichment [32,95,96]. Unfortunately, it can be difficult to extract an intuitive meaning from these measures [97], and they are often not comparable across test sets because their scale and value depends on the set size and number of decoys in the test set. Here, we introduce a simple new early enrichment measure, the *decoy retention factor* (DRF); DRF is easy to interpret, and generalizes across input size. We note that DRF is only applicable in situations in which the number of active and decoy ligands is known beforehand. For the analysis of fingerprint benchmarks, we present both DRF and AUC values. Additional metrics such as BEDROC [94] and sum of log rank [32] are summarized visually in the Appendix A.

The purpose of DRF is to identify, for a parameterized fraction *p* of the active molecules, how effectively decoys are filtered from the score range containing those actives. Consider an input containing *n* active compounds and *d* decoys, and an enrichment threshold of p=0.1. Since we are interested in the score of the top *p* fraction of actives, let x=⌈pn⌉, and let $s_{p}$ be the score of the *x*-th element (so that $s_{p}$ is the score threshold that would recover 10% of actives). Define $d_{p}$ to be the number of decoys that exceed $s_{p}$—this is a fraction of the *d* total decoys. DRF${\;\;}_{p}$ measures the extent to which decoys have been filtered out of the range containing the top *p* actives:(1)$$DRF_{p}=\frac{d_{p}}{\left\lfloor pd\right\rfloor }$$

A system that assigns scores randomly will recover a fraction *p* of the decoys at roughly the same score as it sees *p* of the actives, so that DRF${\;\;}_{p}=1$. In an ideal case, no decoys have score greater than the *x*-th active element, meaning that $d_{p}=0$, and thus DRF${\;\;}_{p}=0$. A DRF${\;\;}_{p}=0.2$ indicates that the number of decoys remaining is 20% of the number expected if *p* of the decoys were kept (there is a 5-fold reduction in decoys). Meanwhile a DRF${\;\;}_{p}>1$ indicates that the method *enriches* for decoys.

We find DRF to be a useful measure because it enables the prediction of the number of decoys expected to remain in a score-filtered result set, based on the size of the underlying library. For example, consider a library of 1 million molecules—this will consist almost entirely of inactives (decoys), so that d≈ 1,000,000. If we hope to discover 10% of actives, and we have previously established that DRF${\;\;}_{0.1}=0.05$ (a 20-fold reduction in decoys relative to random chance), then we expect to observe $d_{p}\;\cdot DRF_{p}\;\approx$ 1,000,000 · 0.1 · 0.05 = 5000 decoys mixed with the surviving actives. This simple calculation is important, because it can highlight that apparently good enrichment (10- or 100-fold) may not be enough to effectively filter out inactives when the target set includes billions of candidates.

## Figures and Tables

**Figure 1 pharmaceuticals-17-00992-f001:**
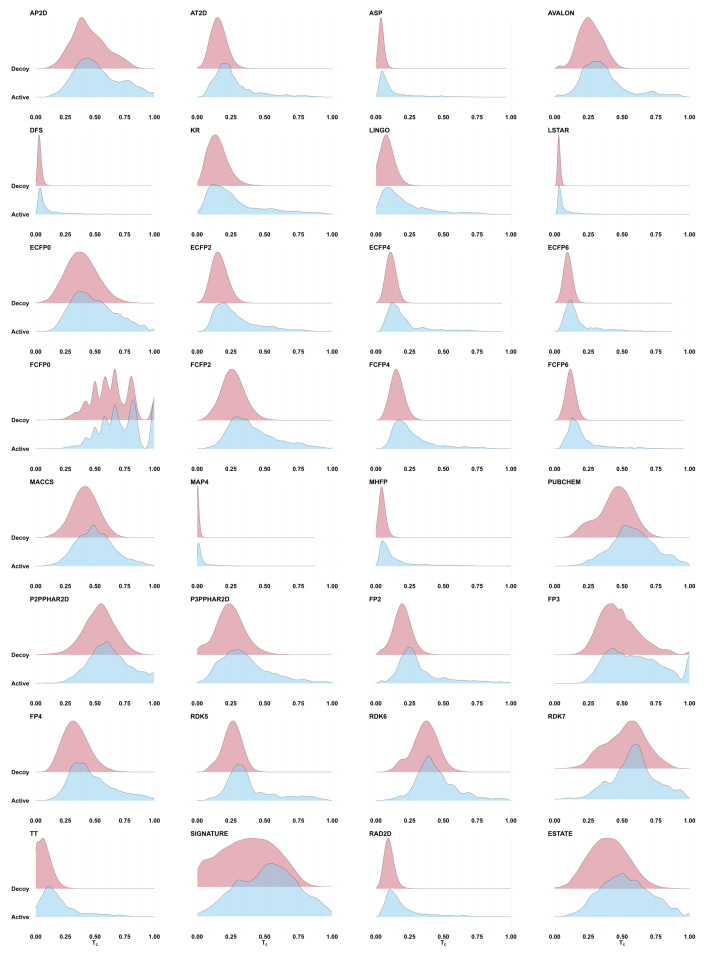
Ridgeline plots showing the distribution of the Tanimoto fingerprint similarities calculated between a randomly selected active molecule for each target protein and all other actives (shown in blue) and decoys (in red) for that target. Data taken from the DEKOIS data set. The distribution of similarity scores between the active query and other active molecules is largely indistinguishable from the distribution of similarity scores to random molecules. Where the active (blue) distribution does show a fatter high-scoring tail than the inactive distribution (suggesting potential for early enrichment by using a high score threshold), a search against a large target database will still produce filtered sets that are massively dominated by inactives (see text).

**Figure 2 pharmaceuticals-17-00992-f002:**
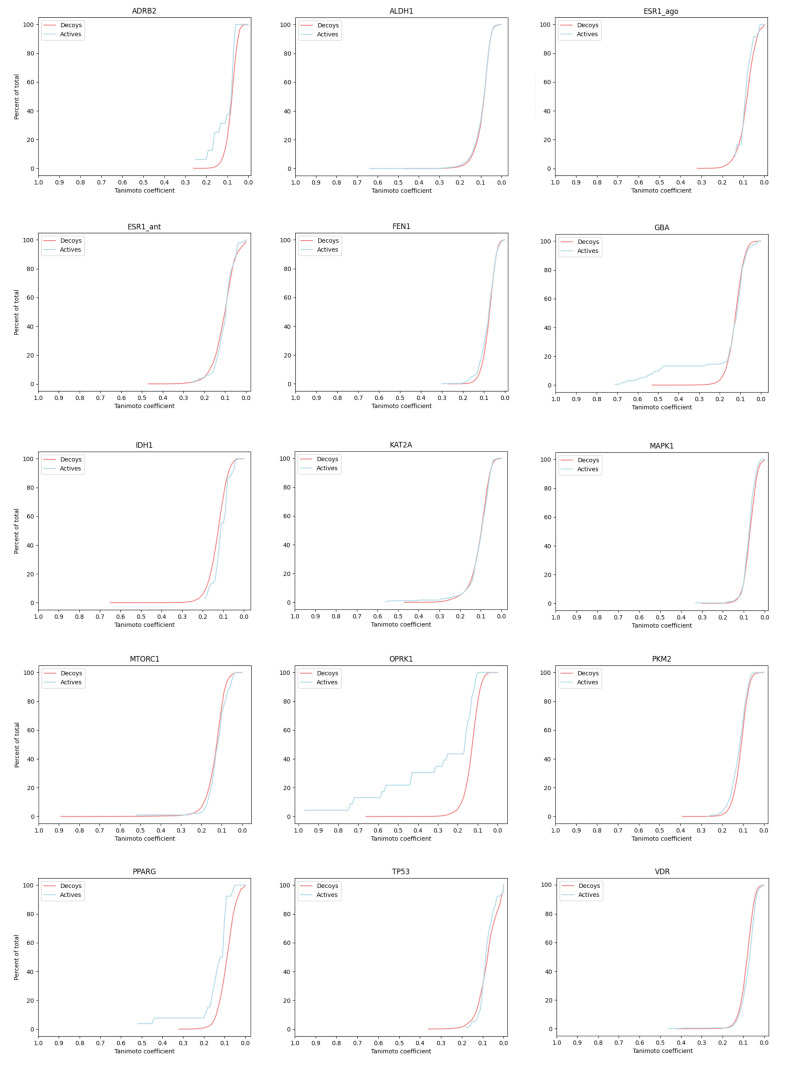
Cumulative percent of total actives (blue) and decoys (red) encountered (Y-axis) as a function of decreasing Tanimoto coefficient, using ECFP4 fingerprints. For each protein, the molecule with the best binding affinity was used as the query molecule (the molecule for which the Tanimoto score was computed for each other molecule, active or inactive).

**Figure 3 pharmaceuticals-17-00992-f003:**
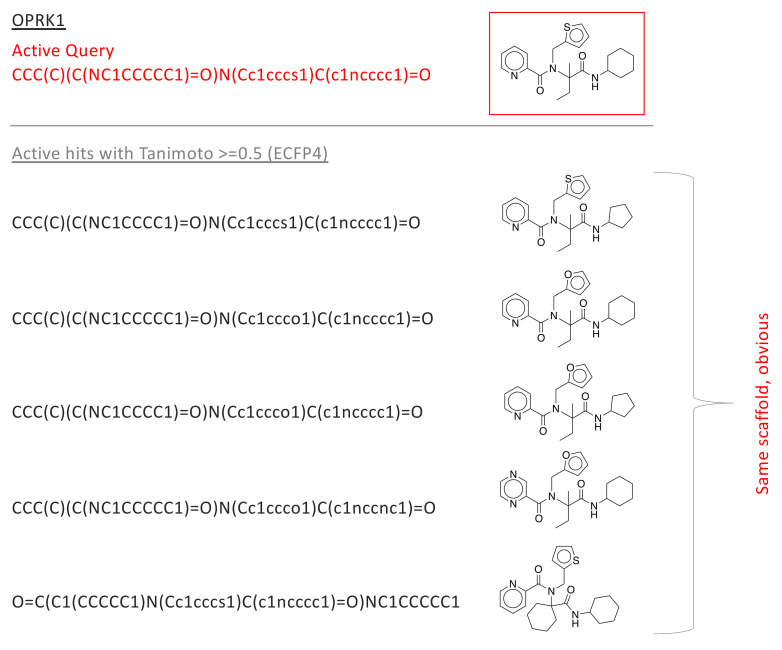
In the LIT-PCBA analysis, early enrichment was observed for OPRK1—for all Tanimoto coefficients t>∼0.2, the fraction of actives with Tanimoto score >t is much larger than the fraction of decoys with that score (see Figure 2). This suggests that fingerprints produce useful early enrichment of active molecules, at least for this one protein target. We sought to understand if the high-scoring actives represented novel drug candidates that could not be easily identified by simple modifications to the active drug used as a fingerprint query. There are only 24 actives in the OPRK1 data set, and only 5 of these showed Tanimoto score >0.5 to the initial query; we manually inspected the structures of these compounds. All 5 are built on the same scaffold as the query (in red), and are obvious variations that should be identified through standard enumeration (i.e., no new scaffold are explored). Similar plots are provided for GBA and PPARG in Appendix A.

**Figure 4 pharmaceuticals-17-00992-f004:**
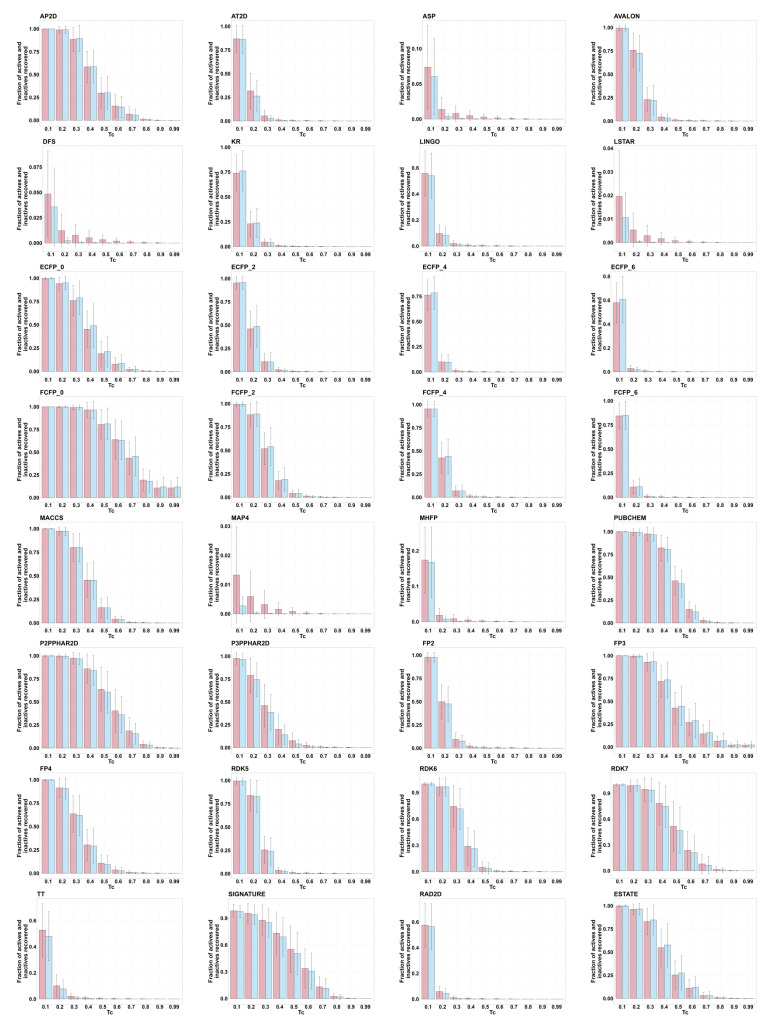
For compounds in the St. Jude malaria data set, bar plots show the fraction of the inactives (in red) and actives (in blue) exceeding Tanimoto similarity cutoffs by the different fingerprints. Tanimoto similarities were calculated using each active as the query; mean and standard deviation (based on the 2507 actives) are shown as error bars.

**Figure 5 pharmaceuticals-17-00992-f005:**
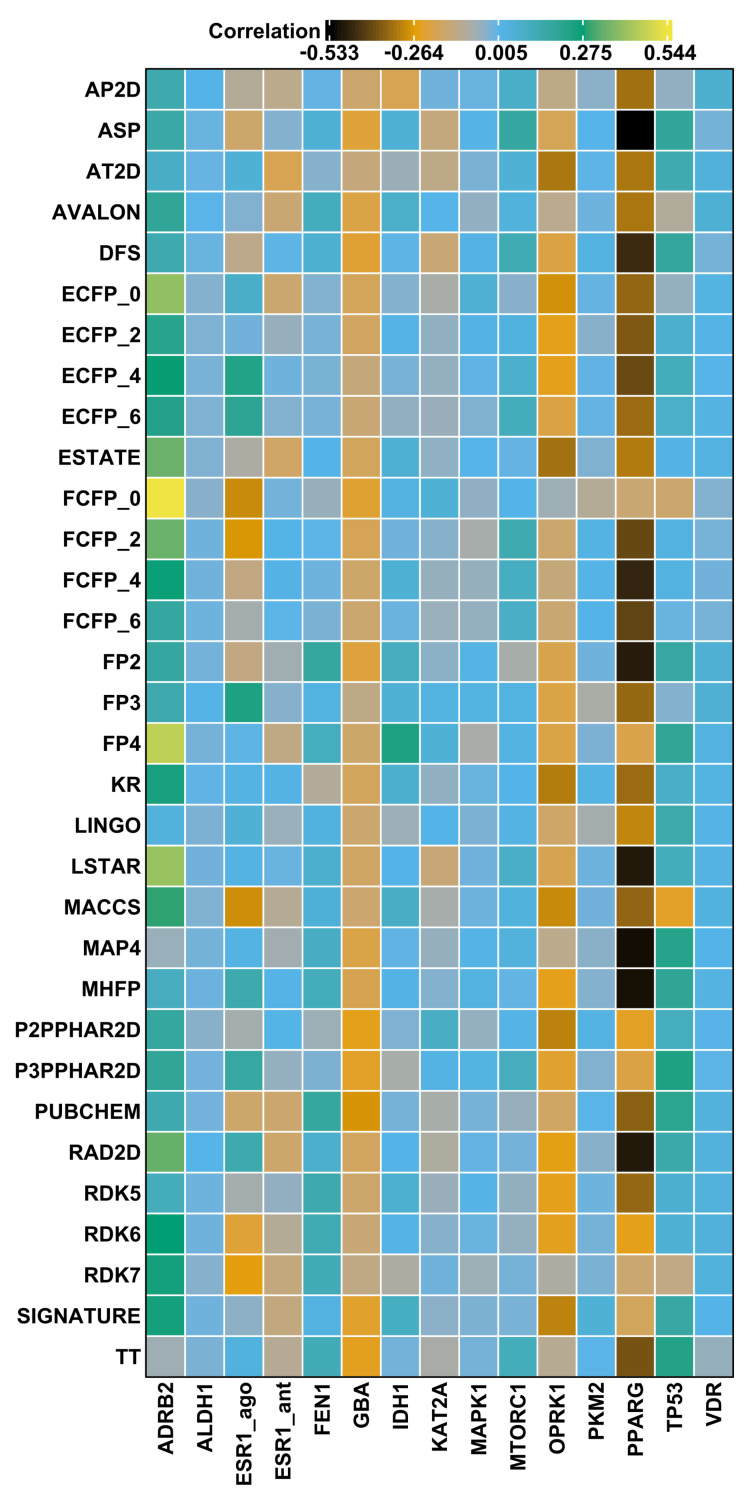
Heatmap of the Kendall rank correlation (τ) between fingerprint Tanimoto ($T_{c}$) similarities calculated between the most active compound for a given target and the potency values (AC${\;\;}_{50}$) of the actives for that target.

**Figure 6 pharmaceuticals-17-00992-f006:**
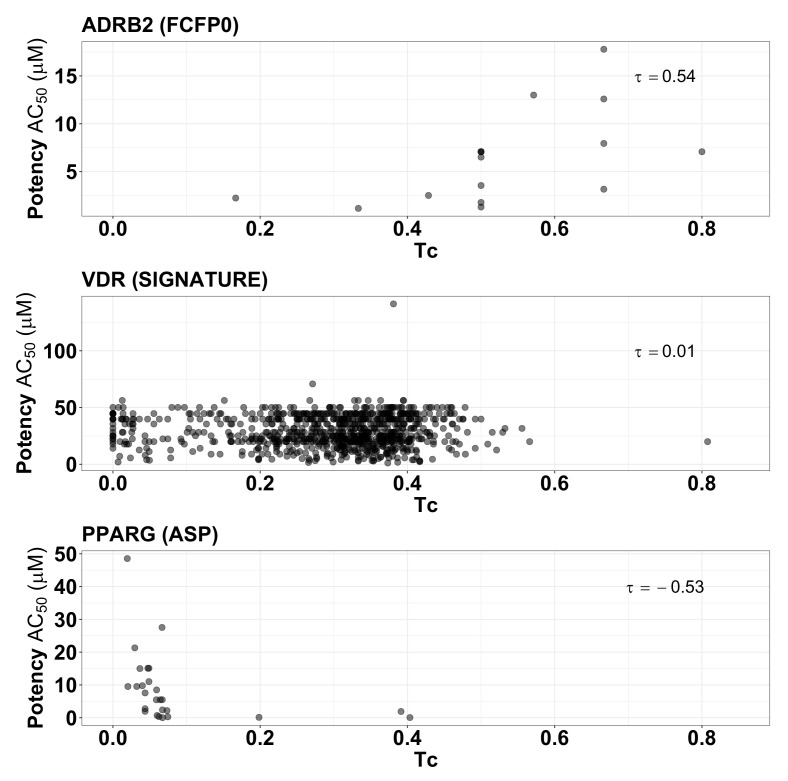
Scatter plot of the fingerprint similarities ($T_{c}$) and the potencies ($AC_{50}$) of active compounds for ADRB2 (using FCFP0 fingerprint), VDR (SIGNATURE fingerprint), and PPARG (ASP fingerprint).

**Table 1 pharmaceuticals-17-00992-t001:** Summary of the VS performances in terms of the AUC and DRF (p=0.1) for the 32 fingerprints tested on the DEKOIS, DUDE, MUV and LIT-PCBA data sets. Each row with no background corresponds to a fingerprint. Each row with a grey background corresponds to the family of fingerprints given in the preceding rows, and shows the mean value for that family. See also heatmaps of the different metrics for the data sets in Appendix A.

FP	AUC				DRF			
	DEKOIS	DUDE	MUV	LIT-PCBA	DEKOIS	DUDE	MUV	LIT-PCBA
AP2D	0.64	0.66	0.49	0.51	0.55	0.39	1.24	1.00
AT2D	0.78	0.79	0.58	0.55	0.20	0.12	0.86	0.83
AVALON	0.72	0.73	0.60	0.55	0.30	0.18	0.85	0.97
ESTATE	0.71	0.75	0.53	0.50	0.31	0.17	0.98	0.94
FP3	0.68	0.77	0.51	0.52	0.45	0.20	0.89	0.83
FP4	0.74	0.80	0.58	0.54	0.30	0.12	0.89	0.91
MACCS	0.71	0.75	0.55	0.54	0.33	0.18	0.99	0.93
PUBCHEM	0.76	0.76	0.55	0.54	0.28	0.20	1.06	0.97
RDK5	0.76	0.75	0.58	0.56	0.24	0.16	0.90	0.89
RDK6	0.70	0.70	0.59	0.58	0.38	0.29	0.86	0.82
RDK7	0.62	0.63	0.58	0.59	0.74	0.70	0.98	0.85
KR	0.72	0.74	0.54	0.51	0.31	0.19	1.05	0.96
SIGNATURE	0.72	0.72	0.55	0.54	0.33	0.23	0.97	0.99
SUBSTRUCTURE	0.71	0.73	0.56	0.54	0.36	0.24	0.96	0.91
ASP	0.80	0.79	0.58	0.53	0.17	0.12	0.89	0.99
DFS	0.79	0.78	0.55	0.52	0.18	0.14	0.98	1.01
FP2	0.79	0.78	0.55	0.54	0.20	0.14	1.01	0.90
LSTAR	0.78	0.78	0.54	0.51	0.19	0.13	0.97	1.03
TT	0.80	0.80	0.61	0.56	0.15	0.10	0.72	0.85
PATH	0.75	0.75	0.57	0.54	0.18	0.13	0.91	0.96
ECFP0	0.70	0.77	0.53	0.50	0.33	0.13	0.97	0.96
ECFP2	0.77	0.81	0.54	0.51	0.19	0.09	0.99	1.01
ECFP4	0.76	0.80	0.54	0.51	0.19	0.09	0.99	1.00
ECFP6	0.75	0.78	0.54	0.52	0.20	0.10	0.99	0.98
FCFP0	0.66	0.69	0.54	0.52	0.35	0.23	0.41	0.42
FCFP2	0.76	0.75	0.55	0.52	0.24	0.19	0.93	1.01
FCFP4	0.78	0.76	0.54	0.52	0.20	0.15	0.93	1.01
FCFP6	0.78	0.75	0.54	0.52	0.20	0.15	0.96	0.98
MAP4	0.81	0.83	0.56	0.54	0.14	0.07	0.91	0.85
MHFP	0.81	0.81	0.54	0.53	0.17	0.10	0.97	0.94
RAD2D	0.76	0.77	0.53	0.53	0.23	0.14	0.99	0.93
CIRCULAR	0.76	0.77	0.54	0.52	0.26	0.11	0.98	0.99
P2PPHAR2D	0.66	0.74	0.51	0.54	0.50	0.25	1.20	0.94
P3PPHAR2D	0.71	0.76	0.52	0.55	0.33	0.16	1.17	0.93
PHARMACOPHORE	0.75	0.76	0.54	0.53	0.42	0.21	1.19	0.94
LINGO	0.77	0.79	0.54	0.54	0.21	0.10	1.02	0.91

**Table 2 pharmaceuticals-17-00992-t002:** LIT-PCBA provides sets of confirmed actives and inactives for 15 protein binding partners.

Protein	Actives	Inactives
ADRB2	17	311,748
ALDH1	5363	101,874
ESR_ago	13	4,378
ESR_antago	88	3,820
FEN1	360	350,718
GBA	163	291,241
IDH1	39	358,757
KAT2A	194	342,729
MAPK1	308	61,567
MTORC1	97	32,972
OPRK1	24	269,475
PKM2	546	244,679
PPARG	24	4,071
TP53	64	3,345
VDR	655	262,648

**Table 3 pharmaceuticals-17-00992-t003:** Molecular fingerprints evaluated in this study. Abbreviations: Topological torsion (TT), Extended Connectivity Fingerprint (ECFP), Functional Class Fingerprint (FCFP), Atom Pair (AP2D), Atom Triplet (AT2D), All Star Paths (ASP), Depth First Search (DFS).

TYPE	FAMILY	DESCRIPTION	SIZE (bits)
AP2D [66]	SUBSTRUCTURE	Topological Atom Pairs	4096
ASP [67]	PATH	All-Shortest Path encoding	4096
AT2D [67]	SUBSTRUCTURE	Topological Atom Triplets	4096
AVALON	SUBSTRUCTURE	Enumerates paths and feature classes	1024
DFS [68]	PATH	All-path encodings	4096
ECFP_0 [69,70]	CIRCULAR	Extended-connectivity fingerprint of diameter 0	1024
ECFP_2 [69,70]	CIRCULAR	Extended-connectivity fingerprint of diameter 2	1024
ECFP_4 [69,70]	CIRCULAR	Extended-connectivity fingerprint of diameter 4	1024
ECFP_6 [69,70]	CIRCULAR	Extended-connectivity fingerprint of diameter 6	1024
ESTATE [70,71]	SUBSTRUCTURE	Fingerprint based on E-State fragments	79
FCFP_0 [69,70]	CIRCULAR	Feature-class fingerprint of diameter 0	1024
FCFP_2 [69,70]	CIRCULAR	Feature-class fingerprint of diameter 2	1024
FCFP_4 [69,70]	CIRCULAR	Feature-class fingerprint of diameter 4	1024
FCFP_6 [69,70]	CIRCULAR	Feature-class fingerprint of diameter 6	1024
FP2 [72]	PATH	Indexes linear fragments up to 7 atoms in length	–
FP3 [72]	SUBSTRUCTURE	Based on 55 SMARTS patterns defining functional groups	–
FP4 [72]	SUBSTRUCTURE	Based on SMARTS patterns defining functional groups	–
KR [70,73]	SUBSTRUCTURE	Klekota–Roth SMARTS-based fingerprint	4860
LINGO [16,70]	TEXT	Fragmentation of SMILES strings	–
LSTAR [67]	PATH	Local Path Environments	4096
MACCS [74]	SUBSTRUCTURE	Molecular ACCess System structural keys	166
MAP4 [75]	CIRCULAR	Combines substructure and atom-pair concepts	2048
MHFP [76]	CIRCULAR	Encodes circular substructures	2048
P2PPHAR2D [77]	PHARMACOPHORE	Pharmacophore pair encoding	4096
P3PPHAR2D [77]	PHARMACOPHORE	Pharmacophore triplet encoding	4096
PUBCHEM [70,78]	SUBSTRUCTURE	Substructure fingerprint	881
RAD2D [22]	CIRCULAR	Topological Molprint-like fingerprints	4096
RDK5 [79]	SUBSTRUCTURE	Encodes substructures at most 5 bonds long	1024
RDK6 [79]	SUBSTRUCTURE	Encodes substructures at most 6 bonds long	1024
RDK7 [79]	SUBSTRUCTURE	Encodes substructures at most 7 bonds long	1024
SIGNATURE [65,70]	SUBSTRUCTURE	Based on an array of atom signatures	–
TT [66]	PATH	Based on bond paths of four non-hydrogen atoms	–

**Table 4 pharmaceuticals-17-00992-t004:** Comparison between different VS data sets. In all cases, the actives may not bind to the same pocket of the target.

Data Set	Active Source	Decoy Generation	Comments
**DUD-E**	ChEMBL09	Total of 0.65% from experiments. Total of 99.35% generated choosing different topologies with similar chemical properties using 2D similarity methods.	No rigorous method to remove false positives. Decoys biased towards 2D similarity methods.
**DEKOIS**	DUD (from literature) [87]	Decoys generated choosing different topologies with similar chemical properties using 2D similarity methods.	Low active to decoy ratio. Decoys biased towards 2D similarity methods
**MUV**	PubChem BioAssay	Unbiased distribution of decoys from experimentally available data.	Data processed to remove false positives and assay artifacts [88]. Low active-to-decoy ratio.
**LIT-PCBA**	PubChem BioAssay	Decoys were chosen from experimentally available data and pruned to have chemical properties similar to actives.	High active to decoy ratio. Actives may not bind to the same pocket of a target. Variable performance in 2D and 3D similarity search and docking across different target sets. Data processed to remove false positives and assay artifacts [88].

## Data Availability

Data and software used for the calculation of fingerprints and scripts to reproduce the results are available from https://osf.io/d3cbr/.

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
