# Peer review of "Do Molecular Fingerprints Identify Diverse Active Drugs in Large-Scale Virtual Screening? (No)"

_pharmaceuticals, 2024, doi:10.3390/ph17080992_

Round 1

Reviewer 1 Report

Comments and Suggestions for Authors

Dear Editor,

The manuscript “Do molecular fingerprints identify diverse active drugs in large-scale virtual screening? (no)” by Vishwesh Venkatraman et. al., has focused on capacities of computational approaches in small-molecule drug discovery by fingerprint similarities. They discussed accuracy and sensitivity of most applicable methods currently are in use based on similarity searches among the chemical data banks. The idea and the fundamentals of the study have been explained clearly in the introduction. They made a good comparison among the molecular fingerprinting strategies for prediction of similar activity using various virtual screening data sets. The pitfall in the strategies is obviously important to be understood and resolved in pharmacological studies, so the paper would be of interest to the audience of Pharmaceuticals. However, some minor need to be reviewed before further consideration:

1-      The authors introduced the decoy retention factor (DRF); as a factor easy to interpret, and generalize across input size. However, it is only applicable in situations in which the number of active and decoy ligands is known beforehand. This condition is not without problems and contradictions in real studies of drug screening, especially in very large scales.

2-      Page 21 Line 8, the first occurring abbreviation “AC50”, must be accompanied by the full form.  Isn’t “IC50” more generic?

3-      The sentence in page 19 at the bottom of the table, “For each …” needs to be rewritten.

4-      A more suitable and attractive form of the title is recommended.

5-      There are no keywords.

Best regards,

Comments on the Quality of English Language

A brief review of the language is recommended.

Author Response

Comment 1: The authors introduced the decoy retention factor (DRF); as a factor easy to interpret, and generalize across input size. However, it is only applicable in situations in which the number of active and decoy ligands is known beforehand. This condition is not without problems and contradictions in real studies of drug screening, especially in very large scales.

Response 1:  In introducing DRF, we do not suggest that the measure is a useful way to predict method efficacy in a de novo search, just that it is useful in exploring the utility of fingerprints for large-scale screening (which is the purpose of our paper). In order to evaluate the efficacy, we need a list of active and inactive molecules, so that predictions can be evaluated. Thus, the limitation of DRF to "known label" circumstances is reasonable.  (Note: we mention this limitation in the DRF introduction, indicating that it not suitable as an accuracy predictor in a de novo search.

~~~

Comment 2: Page 21 Line 8, the first occurring abbreviation “AC50”, must be accompanied by the full form.  Isn’t “IC50” more generic?

Response 2: We have modified the text of the document to define AC50. Note that we retrieved the values from PubChem, which doens't provide IC50 for the compounds in LIT-PCBA.

~~~

Comment 3:  The sentence in page 19 at the bottom of the table, “For each …” needs to be rewritten.

Response 3: We have modified the text to improve clarity.

~~~
Comment 4: A more suitable and attractive form of the title is recommended.

Response 4: We believe this is a stylistic matter, and have intentionally chosen the given format to rapidly communicate the point of the paper 

~~~

Comment 5: There are no keywords.

Response 5: Keywords were included in our latex document, but were apparently not presented in the pdf. We have updated to the latex template, and the keywords now appear.

Reviewer 2 Report

Comments and Suggestions for Authors

The study analyses the performances of common molecular fingerprints in identifying active molecules in ligand-based virtual screening campaigns and highlights that their overall performances are not so satisfactory. The study is well designed, clear and technically sound. The reported results can be interesting for the computational chemists involved in similarity-based virtual screening analyses. Nevertheless, there are some points that the Authors should address before publication.

1) Introduction: among the problems which can undermine the performances of fingerprints, the reviewer suggests to also include the concept of activity cliffs.

2) Methods: Along with the DRF parameter, the most common enrichments factors should be utilized

3) Results: in the enrichment analysis, a more detailed initial description of how the results for all the active compounds are combined would enhance the understanding

4) Results: Along with the analysis of the different performances for the various utilized datasets, the reviewer suggests to also include a brief description of the different performances for the various implemented fingerprints

4) Results: While in the enrichment analysis all active molecules are considered, in Tanimoto calculations only one randomly chosen active molecule is used as the query. The reviewer believes that the analysis should consider more than only one active molecule to reduce the randomness of the obtained results

5) Results: as mentioned by the Authors, the reviewer agrees that the MMV dataset is not so suitable for this study because it includes molecules with different mechanisms of action.

6) May there be a role in these analyses for the concept of applicability domain? If yes, how it could be implemented?

Author Response

Comment 1: Introduction: among the problems which can undermine the performances of fingerprints, the reviewer suggests to also include the concept of activity cliffs.

Response 1: As suggested, we have added mention of this concept to the introduction. 

~~~

Comment 2: Methods: Along with the DRF parameter, the most common enrichments factors should be utilized

Response 2: The supplementary materials page provides plots showing the (poor) correlation between fingerprint similarity and other common enrichment measures.

~~~

Comment 3: Results: in the enrichment analysis, a more detailed initial description of how the results for all the active compounds are combined would enhance the understanding

Response 3: Done. We've added text to the end of the first paragraph explaining each number reported in the table is an average (mean) of per-target values for the fingerprint-benchmark pair

~~~

Comment 4: Results: Along with the analysis of the different performances for the various utilized datasets, the reviewer suggests to also include a brief description of the different performances for the various implemented fingerprints

Response 4: It's possible that we don't understand the reviewer's request. Table 3 presents and extensive collection of results, showing performance of each of 32 fingerprints on all 4 benchmarks. The text includes a high-level discussion of these results (paragraph 2 of "Enrichment for active molecules"). The big picture: there is a little bit of variability among the fingerprints, but none of them is effective. Futher detailed discussion of the various fingerprints seems unlikely to provide additional information to the reader.

~~~

Comment 5: Results: While in the enrichment analysis all active molecules are considered, in Tanimoto calculations only one randomly chosen active molecule is used as the query. The reviewer believes that the analysis should consider more than only one active molecule to reduce the randomness of the obtained results

Response 5: We have added text to end of the first paragraph explaining the reasoning behind using a single query molecule. Briefly: we believe that this better indicates the ditributions observed when using fingerprints in screening, and avoids visual complications resulting from computing mixtures of distributons. 

~~~

Comment 6: Results: as mentioned by the Authors, the reviewer agrees that the MMV dataset is not so suitable for this study because it includes molecules with different mechanisms of action.

Response 6: We agree (and mention in the manuscript) that MMV is not an ideal benchmark for virtual screening, because the pocket of each active is unknown. Even so, we strongly believe that it is a useful dataset for performing a supplementry analysis, because it provides a massive set of verified inactive molecules. Though imperfect on its own, it provides another perspective on the weakness of fingerprints.

~~~

Comment 7: May there be a role in these analyses for the concept of applicability domain? If yes, how it could be implemented?

Response 7: We have added text to the Discussion in which we discuss that molecular fingerprints are likely to be unhelpful in estimating classification confidence or applicability domain.

Reviewer 3 Report

Comments and Suggestions for Authors

In my opinion, this paper could be worthy of publication by a much more theoretical journal than Pharmaceuticals, I think  it’s not suitable for Pharmaceuticals. The proposed strategy to better rank active/inactives should be accompanied by screening and experimental evaluation of novel compounds, to be this paper considered in medicinal chemistry journals. The choice of the reported fingerprints could be not so effective to rank the benchmarking dataset compounds. These set of molecules are differently conceived and include a very high number of putative inactive compounds (rather than actives).; Activity and inactivity should be related to binding events or functional events, being determined or not in different ways. This could impair predictive calculations.

Minor comments:

Introduction

The sentence (hereafter referred to as actives) should be moved where it has been used for the first time.

A workflow of the whole study should be added.

Methods and materials

Please revise Table 1 listing each type based for family.

The sentence “Some benchmarks include only confirmed inactive molecules, while others add

compounds presumed to be non-binding” should be more detailed adding information of previous studies.

Table 2 should be moved prior to DUD-E explanation. The first comment Field is unclear. I do not understand if you state a very general comment on the dataset or if you explain your evaluation procedure (Removal of false positives and assay artifacts).

Labels in Figures 1, 2 and 4 are difficult to read, please revise them.

Caption to Figure 1 should be simplified and detailed in the main text.

Author Response

Comment 1: The sentence (hereafter referred to as actives) should be moved where it has been used for the first time.

Response 1: We have revised the text to define "actives" where it first appears in the Introduction.

~~~

Comment 2: A workflow of the whole study should be added.

Response 2: This mansucript describes a collection of experiments that each show the futility of using molecular fingerprints in the search of novel active molecules for a given protein targets. The datasets used differ in ways that allow verious perspectives on the weakness of fingerprints. There is no comprehensive workflow over the entire paper - it's a series of analyses, each described in its own section. 

~~~

Comment 3: Please revise Table 1 listing each type based for family.

Response 3: We think this is a request to reorder the list of fingerprints so that they are grouped by Family. We respectfully suggest that this table is best represented in complete alphabetical order (and note that the fingerprints are sorted by family type in Table 3). If the Editors insist, we are willing to change the order of Table 1.

~~~

Comment 4: The sentence “Some benchmarks include only confirmed inactive molecules, while others add compounds presumed to be non-binding” should be more detailed adding information of previous studies.

Response 4: This information is provided in the benchmark descriptions found in Table 2 and the descriptive text surrounding it. 

~~~

Comment 5: Table 2 should be moved prior to DUD-E explanation. The first comment Field is unclear. I do not understand if you state a very general comment on the dataset or if you explain your evaluation procedure (Removal of false positives and assay artifacts).

Response 5: When we adjusted to the MDPI LaTeX format, the table moved to a position that agrees with this request. We have adjusted the Comments for clarity.

~~~

Comment 6: Labels in Figures 1, 2 and 4 are difficult to read, please revise them.

Response 6: We have adjusted these label font sizes

~~~
Comment 7: Caption to Figure 1 should be simplified and detailed in the main text.

Response 7: This seems to be a different in communication style (i.e. a stylistic choice). We find it helpful to provide enough details in the figure caption to allow the reader to understand the figure without needing to hunt through the main body of the text.

Reviewer 4 Report

Comments and Suggestions for Authors

Venkatraman et al. presents a thorough and critical evaluation of molecular fingerprints in virtual screening, demonstrating their limitations in identifying diverse active compounds.

The study provides an evaluation of 32  molecular fingerprints across multiple benchmark datasets (DUD-E, MUV, DEKOIS, LIT-PCBA). The authors introduce the Decoy Retention Factor (DRF) which offers a novel and practical approach to assessing early enrichment in virtual screening, addressing some limitations of traditional metrics like AUC. The authors identify significant limitations in using molecular fingerprints for virtual screening, particularly their inability to distinguish between active and inactive molecules effectively and their poor correlation with compound potency.

Suggestions and comments

1.     While the authors discuss the general inadequacy of molecular fingerprints, a more detailed comparative analysis of which specific fingerprints performed better or worse could provide valuable insights

2.     The study notes potential biases in benchmarking datasets but lacks a detailed discussion on their impact and mitigation. Further exploration of these biases is needed.

3.     The authors suggest moving beyond fingerprints, mentioning surface property representations. Expand on future directions with examples like geometric deep learning and surface patch analysis.

 My recommendation is minor revisions.

Author Response

Comment 1: While the authors discuss the general inadequacy of molecular fingerprints, a more detailed comparative analysis of which specific fingerprints performed better or worse could provide valuable insights

Response 1: The key point of this manuscript is that none of the molecular fingerprints is useful for identifying novel active drugs for a given target. We do comment on the performance of ECFP2 in the text for Figure 1, as it appears to be the best of the bunch in this analysis -- we show that even this relatively-good performance is not particularly useful. We also explore the relative performance of MAP4 in the MMV analysis, again showing that its performance is lacking. We believe there is little value in further exploring the extent to which one approach is somewhat less terrible than another approach.

~~~

Comment 2: The study notes potential biases in benchmarking datasets but lacks a detailed discussion on their impact and mitigation. Further exploration of these biases is needed.

Response 2: Biases in the benchmarks have been discussed at length in other articles (for example references 53,55,72–77 as cited in our manuscript). We agree that it is important to better understand these biases, and to develop improved benchmarks that mitigate these problems, but hold that further exploration is outside the scope of this paper (which is about fingerprints).  

Note that, in  our discussion of Table 3, we observed that the DUDE and DEKOIS decoys were selected by picking decoys with low fingerprint similarity to the actives, so that fingerprints should by design be effective ... and that even with this design, fingerprints are not particularly useful. 

~~~

Comment 3: The authors suggest moving beyond fingerprints, mentioning surface property representations. Expand on future directions with examples like geometric deep learning and surface patch analysis.

Response 3: The final paragraph of the discussion addresses potential paths forward based on deep learning and surface patch analysis. We believe further discussion (such as describing potential solutions in detail) are outside the scope of this project.

Round 2

Reviewer 3 Report

Comments and Suggestions for Authors

I still think  this manuscript is not suitable for Pharmaceuticals. The proposed strategy to better rank active/inactives should be accompanied by screening and experimental evaluation of novel compounds, to be this paper considered in medicinal chemistry journals.

Author Response

Comment 1

“The proposed strategy to better rank active/inactives should be accompanied by screening and experimental evaluation of novel compounds, to be this paper considered in medicinal chemistry journals. The choice of the reported fingerprints could be not so effective to rank the benchmarking dataset compounds. These set of molecules are differently conceived and include a very high number of putative inactive compounds (rather than actives).; Activity and inactivity should be related to binding events or functional events, being determined or not in different ways. This could impair predictive calculations.”

Response 1

We reject the premise that this paper would be improved by performing a new screening and experimental evaluation of novel compounds. This threshold for publication is imaginary, as demonstrated by examples of papers that this journal has published in the recent past with computational approaches applied to existing benchmarks without producing new data:

  • https://doi.org/10.3390/ph15060691
  • https://doi.org/10.3390/ph15050613
  • https://doi.org/10.3390/ph14080790
  • https://doi.org/10.3390/ph14080758

Moreover, our manuscript demonstrates the weakness of a commonly-used computational method via application of the method to a diverse pool of benchmarks that have been carefully constructed by others for the explicit purpose of measuring computational prediction of binding activity. Each of those benchmarks has some flaws/biases (documented in our manuscript and the literature), so we considered the collection, rather than only a single one of them, to show that the performance of molecular fingerprints is poor under all angles of inquiry that we can test (i.e. failure is not just due to some single benchmark). Importantly, there are many compounds in the datasets, so that our evaluation is much more diverse than it would be in an evaluation based on a single novel compound. Note that we supplemented the evaluation with a large experimentally validated dataset, which is effectively equivalent to the new evaluation called for by Reviewer #3.

We have added text to the end of the first paragraph of “Benchmarking Data Sets” to highlight the fact that using multiple benchmarks, each with distinct flaws, provides many perspectives to our evaluation, indicating that limited ability to differentiate decoys from active molecules is not simply due to a specific design flaw found in a single benchmark.

~~~~~

Comment 2

“… please make the main scope and objectives of your work more clear and detailed in the Introduction and Materials and Methods section.”

Response 2

It’s not clear what improvements the reviewer imagines should be made. We have reviewed the entire manuscript, and believe the scope and objectives are already clearly laid out in the Introduction and M&M sections.

~~~~~

Comment 3

The Discussion section is too short and should be expanded….

Response 3

We have added a relevant paragraph to the Discussion section.

Round 3

Reviewer 3 Report

Comments and Suggestions for Authors

In my opinion this manuscript is still too speculative, poorly validated and not suitable for Pharmaceuticals.